# Composite Magnetic Filaments: From Fabrication to Magnetic Hyperthermia Application

**DOI:** 10.3390/mi16030328

**Published:** 2025-03-12

**Authors:** Athanasios Alexandridis, Apostolos Argyros, Pavlos Kyriazopoulos, Ioannis Genitseftsis, Nikiforos Okkalidis, Nikolaos Michailidis, Makis Angelakeris, Antonios Makridis

**Affiliations:** 1Department of Condensed Matter and Materials Physics, Aristotle University of Thessaloniki, 54124 Thessaloniki, Greece; ath.alexandrid@gmail.com (A.A.); paulkyriazo@gmail.com (P.K.); igenitse@auth.gr (I.G.); agelaker@auth.gr (M.A.); 2Laboratory of Magnetic Nanostructure Characterization, Technology and Applications (MagnaCharta), Centre for Interdisciplinary Research and Innovation, Balkan Centre, Building B’, 10th km Thessaloniki-Thermi Road, 57001 Thessaloniki, Greece; 3Physical Metallurgy Laboratory, Mechanical Engineering Department, School of Engineering, Aristotle University of Thessaloniki, 54124 Thessaloniki, Greece; aargyros@auth.gr (A.A.); nmichail@meng.auth.gr (N.M.); 4Centre for Research & Development of Advanced Materials (CERDAM), Centre for Interdisciplinary Research and Innovation, Balkan Centre, Building B’, 10th km Thessaloniki-Thermi Road, 57001 Thessaloniki, Greece; 5Medical Physics & Digital Innovation Laboratory, School of Medicine, Faculty of Health Sciences, Aristotle University of Thessaloniki, AHEPA University Hospital, 54636 Thessaloniki, Greece; nikiforos_ok@hotmail.com; 6Morphé, Lagkada 33, 54629 Thessaloniki, Greece

**Keywords:** composite magnetic filaments, bone tissue magnetic scaffolds, 4D printing, magnetic hyperthermia, nanocomposite materials

## Abstract

The printing of composite magnetic filaments using additive manufacturing techniques has emerged as a promising approach for biomedical applications, particularly in bone tissue engineering and magnetic hyperthermia treatments. This study focuses on the synthesis of nanocomposite ferromagnetic filaments and the fabrication of bone tissue scaffolds with time-dependent properties. Three classes of polylactic acid-based biocompatible polymers—EasyFil, Tough and Premium—were combined with magnetite nanoparticles (Fe_3_O_4_) at concentrations of 10 wt% and 20 wt%. Extruded filaments were evaluated for microstructural integrity, printed dog-bone-shaped specimens were tested for elongation and mechanical properties, and cylindrical scaffolds were analyzed for magnetic hyperthermia performance. The tensile strength of EasyFil polylactic acid decreased from 1834 MPa (0 wt% Fe_3_O_4_) to 1130 MPa (−38%) at 20 wt% Fe_3_O_4_, while Premium polylactic acid showed a more moderate reduction from 1800 MPa to 1567 MPa (−13%). The elongation at break was reduced across all samples, with the highest decrease observed in EasyFil polylactic acid (from 42% to 26%, −38%). Magnetic hyperthermia performance, measured by the specific absorption rate, demonstrated that the 20 wt% Fe_3_O_4_ scaffolds achieved specific absorption rate values of 2–7.5 W/g, depending on polymer type. Our results show that by carefully selecting the right thermoplastic material, we can balance both mechanical integrity and thermal efficiency. Among the tested materials, Tough polylactic acid composites demonstrated the most promising potential for magnetic hyperthermia applications, providing optimal heating performance without significantly compromising scaffold strength. These findings offer critical insights into designing magnetic scaffolds optimized for tissue regeneration and hyperthermia-based therapies.

## 1. Introduction

In the past decade, 3-dimensional (3D) printing research has intensified, as it is one of the most versatile material synthesis techniques across all scientific disciplines, with numerous important applications [1,2,3,4]. Concurrently, the exploration of “bio” materials [5,6] has surged, driven by their applications in biomedical technology, disease treatment, and regenerative medicine, particularly as scaffolds for bone tissue engineering. This increasing interest necessitates a deeper understanding of the mechanisms involved, alongside the development of new materials and structures that can address these critical challenges.

The use of nanoparticles in medical applications has been well established for decades [7,8]. However, the incorporation of nanostructures into polymer matrices to form nanocomposites for applications in the aforementioned fields remains a relatively novel and promising area of research [9,10].

Magnetism, in particular, plays a crucial role in both biological and technological contexts. Magnetic materials, especially magnetic nanoparticles (MNPs), have been widely studied for their unique properties, such as favorable biocompatibility and distinct magnetic behavior with specific focus on drug delivery, magnetic hyperthermia, MRI, and biological separations [11,12]. Under an external magnetic field, MNPs can adhere to cell membranes, be internalized through endocytosis, and influence cellular functions. In bone tissue engineering, they become magnetized and enable scaffolds to enhance tissue regeneration, combining their magnetic properties with therapeutic benefits. MNPs, with or without therapeutic agents, are increasingly utilized in medical applications, proving especially effective in bone repair [13]. Recently, there has been a growing trend to integrate magnetism into 3D printing technology by incorporating magnetic materials into polymers to create composite filaments [14,15,16,17]. These magnetic polymer composites open new perspectives in biomedical applications, where tailor-made in design, morphology and properties 3D-printed scaffolds may be implemented in tissue regeneration, targeted drug delivery, and as heating agents in magnetic hyperthermia. The ability to tune both the magnetic and structural properties through 3D printing advances bone repair in beneficial combinatory action as localized heat generation thermal seeds under external magnetic fields. Within this framework, this study establishes methods and protocols for designing and printing scaffolds with dynamic, time-evolving properties leading to 4-dimensional (4D) printing. These scaffolds are analyzed in terms of their microstructure, mechanical behavior, and thermal performance. The design, the fabrication methods, and the heating evaluations are based on our previous works, where we developed protocols to fabricate magnetic filaments [15] for use in 3D printing technology to construct magnetic scaffolds and evaluate their efficiency in magnetic hyperthermia for biomedical applications [18,19,20]. Furthermore, considerable effort was dedicated in understanding the structure–property relationships to predict the thermal performance of the scaffolds more accurately during design.

During the design phase, optimal printing parameters were determined to ensure high infill density without compromising the mechanical integrity of the structures, while also maintaining a porosity level similar to that of natural bone tissue. Three classes of chemically modified biocompatible polymers derived from polylactic acid (PLA) (EasyFil PLA, Premium PLA, Tough PLA) were chosen for this purpose, along with two structure types (a cylindrical shape for magnetic hyperthermia measurements and a dog bone shape for tensile mechanical evaluation). Two concentrations of magnetite nanoparticles (10 and 20 wt%) were tested, and all structures were compared to their non-magnetic counterparts to observe changes in microstructure, mechanical performance, and magnetic hyperthermia behavior. The three different PLA modifications allowed us to study the influence of polymer chain configurations on the final properties and to identify the most suitable form for further research and applications. It is important to note that this study takes a novel approach to 4D printing by focusing not on shape memory but on the temperature change during therapeutic magnetic hyperthermia. Previous works have established protocols for fabricating magnetic polymer filaments and demonstrated their heating efficiency for biomedical applications, particularly in magnetic hyperthermia [15,18,19,20]. However, none of these studies have directly correlated the filament fabrication process with the mechanical and thermal performance of the final scaffolds in a systematic manner. Here, we build upon our previous research by refining the filament fabrication protocol and linking it directly to scaffold efficiency in magnetic hyperthermia and mechanical behavior. Furthermore, for the first time, we introduce an alternative protocol for mechanical property evaluation, which has not been previously documented in the literature. This novel approach provides a comprehensive understanding of the interplay between material composition, processing parameters, and functional properties, paving the way for optimized 3D-printed magnetic scaffolds with enhanced biomedical applicability.

## 2. Materials and Methods

### 2.1. Materials

Three types of polylactide (PLA) filaments (1.75 mm, FormFutura [21]) were selected based on their mechanical properties:

EasyFil™ PLA: modified with an impact modifier for improved printability and stability.

Tough™ PLA: high-impact PLA with ABS-like strength but greater stiffness and excellent layer adhesion.

Premium™ PLA: high-purity PLA with superior thermal stability and crystallization speed.

PLA was chosen for its availability, biodegradability, and suitability for biomedical applications due to its non-toxic nature and low melting point (170–180 °C).

Magnetite nanoparticles (MNPs, 50–100 nm, Alfa Aesar, Karlsruhe, Germany) were incorporated as the magnetic phase in composite filaments with 10 and 20 wt% MNP content.

### 2.2. Methods

This section describes the categories of materials and methods used in this study, divided into two main categories: (1) filaments and (2) 3D printouts. Each category is analyzed with the relevant subcategories and techniques used.

#### 2.2.1. Composite Filaments

The composite filaments were fabricated following the protocol described in our previous work [15], which included four main stages: (a) pulverization of commercial filaments, (b) mixing of materials, (c) drying of the composite material, and (d) extrusion of the composite filament.

(A) Pulverization: commercial PLA filaments (EasyFil, Tough, Premium) were manually cut and milled into fine powder, with intermittent freezing below 0 °C to enhance particle uniformity.

(B) Blending: the PLA powder was mixed with Fe_3_O_4_ nanoparticles at 10 wt% (8 g MNPs + 72 g PLA) and 20 wt% (16 g MNPs + 64 g PLA) compositions.

(C) Drying: the composite was dried at 70 °C for 1 h to remove moisture before extrusion.

(D) Extrusion: the dried blend was processed using a VT110 Filament Extruder (3D Tech Printing, Thessaloniki, Greece) at optimized temperatures (e.g., 165 °C for EasyFil PLA, slightly higher for others). The presence of Fe_3_O_4_ required an additional 5–10 °C to compensate for increased viscosity.

For the structural and magnetic characterization of the filaments utilized in this study, several advanced techniques were employed to ensure comprehensive analysis.

Phase identification of the filaments was carried out through X-ray diffraction (XRD) experiments using a SIEMENS D500 X-ray diffractometer (Siemens, Berlin, Germany) with a Cu Kα radiation source. The diffraction data were collected over a 2θ range of 10–80° at a scanning rate of 8° min⁻^1^, enabling detailed phase analysis of the composite materials.

#### 2.2.2. 3D Printouts

##### Mechanical Tests

Tensile properties of the fabricated filaments were assessed using ASTM D638-14 type V standard [22] with dog bone specimens. Testing was conducted on a TA Instrument Electroforce 3550 (Eden Prairie, MN, USA) universal testing machine at 5 mm/min until failure, recording stress–strain curves to extract key mechanical parameters: elastic modulus, tensile strength, and elongation at break.

To ensure consistency and reliability, all tests were conducted under controlled environmental conditions, maintaining a constant temperature and humidity. This eliminated external factors that could influence the mechanical performance of the filaments and ensured repeatability across all experiments. Given the complexity of producing ferromagnetic filaments and the challenges associated with printing custom materials, the ASTM standard was selected for its suitability for small sample sizes. Additionally, the short printing time per specimen, approximately five minutes, allowed for efficient large-scale production. Each dog bone specimen had a total length of 63.5 mm, with an effective gauge length of 9.53 mm. To ensure precision in the final mechanical results, at least three specimens were printed and tested per category (as shown in Appendix A), allowing for statistical analysis, including standard deviation calculations.

Ferromagnetic filaments with different concentrations of magnetic nanoparticles (MNPs) were used to fabricate the dog bone specimens. Specifically, three specimens were printed for each filament type: Premium PLA (Pr. PLA), EasyFil PLA (EF. PLA), and Tough PLA (T. PLA), each containing 10 wt% and 20 wt% of MNPs, resulting in six categories of magnetic samples. Additionally, three non-magnetic control specimens were printed from pure Pr. PLA, EF. PLA, and T. PLA filaments, bringing the total number of filament categories to nine. This comprehensive selection allowed for a direct comparison of the mechanical properties between magnetic and non-magnetic samples.

Extensive research highlights the significant impact of 3D printing parameters, such as layer height, orientation angle, and print speed, on tensile properties like elastic modulus, tensile strength, and elongation at break [23,24]. Higher tensile strength is generally associated with thicker layers and slower print speeds, while the orientation angle primarily affects elastic modulus and elongation at break [25]. Based on these insights, the printing parameters were carefully optimized in this study. The dog bone specimens were printed with a layer height of 0.2 mm, a print speed of 50 mm/s, and a raster angle of 0°, as this configuration aligned the printed layers with the mechanical load direction. A raster angle of 90° was found to produce the lowest tensile strength, while a 45° angle resulted in the weakest overall performance due to fiber misalignment in relation to the applied force.

In Fused Deposition Modeling (FDM), material selection and process parameters significantly influence mechanical properties. Key factors include build orientation, thermal conditions, and slicing parameters [23]. Horizontal, vertical, and lateral build orientations, along with extrusion and bed temperatures, affect part shrinkage and interlayer adhesion. A balance between layer thickness and infill density is crucial, as higher infill densities improve strength but increase print time and material costs.

Minimizing voids enhances mechanical properties, with raster angle, build orientation, and layer thickness playing critical roles [26]. Thinner layers improve interlayer adhesion by reducing void size. To optimize mechanical performance, dog bone specimens were printed with a 1.2 mm wall thickness, a single wall line count, and 80% infill density. A “lines” infill pattern ensured effective load propagation, with a 0/90° raster angle for structural stability. Extrusion temperatures were 200 °C for Premium and EasyFil PLA and 210 °C for Tough PLA, with a build plate temperature of 60 °C. A print speed of 50 mm/s was maintained, and a brim was applied for adhesion.

This methodology ensured reproducible, high-quality fabrication of ferromagnetic nanocomposite filaments and dog bone specimens, suitable for biomedical applications like tissue engineering and magnetic hyperthermia. The dimensions of the fabricated scaffolds and dog bone specimens, including height (H), diameter (D), length (L), and gauge (G), are summarized in Table 1, providing a crucial reference for assessing structural variations and the impact of MNP concentration on mechanical properties.

##### Magnetic Hyperthermia

The magnetic scaffolds were fabricated using custom-made 3D printing filaments and a low-cost FDM printer (LK4 Longer), with designs created in TinkerCad 3D Design software. Scaffold dimensions were consistent across samples, with cylindrical structures measuring 16 mm in height and 19 mm in diameter (Appendix A). Fabrication followed previously established protocols [15,20] for heating efficiency and scaffold preparation, utilizing varying concentrations of magnetic nanoparticles (MNPs) embedded in an agarose matrix. Consistent printing parameters included extrusion temperatures of 200 °C to 210 °C, a build plate temperature of 60 °C, and a print speed of 50 mm/s, ensuring reproducibility across all samples.

For magnetic hyperthermia experiments, the same general approach as that used for fabricating the 3D-printed scaffolds in our previous works was followed [15,20]. The scaffolds were subjected to an alternating magnetic field using the Easyheat AC field induction heating system provided by Ambrell Co. (New York, NY, USA). The system operated at 15.9 kA/m and 400 kHz, with scaffolds embedded in an agarose matrix entered in the induction coil. The coil was cooled using a closed-loop water system, maintaining a temperature of ~17 °C. Temperature measurements were recorded using optical fiber positioned at the center of each scaffold, with a set initial temperature of 17 °C. The magnetic field was applied for 50–150 s, followed by a 2 min cooling period, with temperatures carefully monitored to avoid exceeding 80 °C, ensuring accurate heating response measurement.

## 3. Results

### 3.1. Filament Features

The initial step in evaluating the composition and microstructure of commercial materials and synthetic ferromagnetic filaments involved X-ray diffraction (XRD) analysis. The employed methodology and protocols for synthesizing the ferromagnetic filaments demonstrated consistent repeatability, ensuring that the predetermined mixing ratios were maintained throughout the synthesis of the filaments.

Figure 1 illustrates the X-ray diffraction patterns for various polylactic acid (PLA) materials, including Tough PLA, EasyFil PLA, and Premium PLA, with and without magnetite magnetic nanoparticles (MNPs) 10 and 20 wt%. The diffraction pattern of magnetite (Fe_3_O_4_) is provided as a reference (pink color (bottom curve) corresponds to PDF#19-0629 of magnetite with relative XRD intensities) in Figure 1. The characteristic broad diffraction peak of PLA filaments (Pr. PLA, EF. PLA and T. PLA samples) observed at 19.9° indicates the presence of a semi-crystalline structure. This peak, typically associated with the crystalline regions of PLA, suggests that the material is not entirely amorphous but contains both crystalline and amorphous domains. The semi-crystalline nature of PLA plays a crucial role in its mechanical properties, as it will be demonstrated later through the mechanical analysis of the dog bone specimens. This semi-crystalline structure provides a balance between stiffness and flexibility, which is essential for its application in scaffold fabrication. The broadness of the peak at 19.9° further confirms the partial crystallinity within the PLA matrix. Sharp diffraction peaks in the patterns signify the crystalline phase of magnetite, in the case of composite filaments, whereas the broad regions indicate the semi-crystalline nature of the polylactic acid. Notably, shifts in the diffraction peaks of the polymer–magnetite composites, relative to those of pure magnetite, suggest modifications in the lattice structure of the magnetite nanoparticles. These alterations may result from the thermal treatment applied during synthesis, where temperatures reached approximately 200 °C. Such conditions can induce interfacial stress or strain between the polymer matrix and the magnetite nanoparticles due to differences in thermal expansion coefficients. Additionally, elevated temperatures may enhance chemical interactions at the polymer–magnetite interface, potentially influencing the lattice parameters of the magnetite. The peak intensity in the diffraction patterns serves as a secondary characteristic correlated with nanoparticle concentration. An increase in peak intensity was observed with higher magnetite content, further validating the efficacy of the synthesis protocol implemented in this study [27]. Compared to the pristine PLA filaments, the characteristic peak of PLA in the composite filaments exhibited a significant reduction in intensity or was even absent in some cases, particularly in all PLA types mixed with 20 wt% of magnetite nanoparticles. This phenomenon is likely due to the dominant diffraction peaks of Fe_3_O_4_, which overshadows the PLA peaks, making it less visible in the composite samples. The strong presence of Fe_3_O_4_ diffraction peaks highlights the successful incorporation of magnetic nanoparticles into the PLA matrix, significantly influencing the material’s overall diffraction pattern.

### 3.2. Mechanical Evaluation

Mechanical properties are crucial for bone scaffolds, as they provide essential support during bone repair. Tensile testing of the synthesized filaments plays a key role in evaluating their potential for biomedical applications. A comparison of the stress–strain curves for EF.PLA, T.PLA, and PR.PLA, illustrated in the diagram below, clearly highlights the differences in the mechanical properties of these materials. As shown in Figure 2a, the stress–strain curves indicate that the modulus of elasticity (E) for the three materials does not significantly differ, aligning with both our experimental measurements and the manufacturer’s specifications. The small standard deviation values are also presented in the accompanying bar graph in Figure 2b. Additionally, Figure 2b includes all the mechanical properties derived from the stress–strain curves, such as the modulus of elasticity (E), ultimate tensile strength (σ_UTS_), yield strength (σ_y_), fracture strength (σ_f_), and strain at break (ε).

Several conclusions can be drawn regarding the mechanical behavior of each material, allowing for an interpretation of the property–structure relationship based on the shapes of the curves. T.PLA exhibits the lowest modulus of elasticity, the lowest stress values, and the highest elongation at break, as is shown clearly in Figure 2c. This observation is consistent with the energy absorption measurements reported by the manufacturer, which identifies T.PLA as having the highest impact resistance among all tested polylactic acid polymers [28].

In contrast, EF. PLA displays an alternative behavior, showing a slightly greater degree of elasticity than T. PLA, with its elasticity values falling between those of T. PLA and PR. PLA. While EF. PLA achieves higher maximum tensile strength than T. PLA, the difference is negligible when compared to Pr. PLA. In terms of necking before fracture, EF. PLA demonstrates intermediate behavior relative to the other two materials [29].

Pr. PLA, with the highest modulus of elasticity and stress values, exhibits the most brittle behavior among the materials tested, failing shortly after the yield point. The observed differences in mechanical properties can be attributed to the microstructure and ratios of the isomers of polylactic acid. Literature suggests that stereocomplexation of PLA enantiomers leads to enhanced mechanical properties due to the formation of stereocomplex crystallites with intermolecular cross-links [30] and, consequently, the behavior of Pr. PLA can be interpreted through the shape of its stress–strain curve, which resembles that of cross-linked polymer. This characteristic limits the relative movement of polymer chains, forming stress concentration points that ultimately lead to material failure and an increase in brittleness [31].

It is important to note that a direct quantitative comparison between our experimental measurements and those provided by the manufacturer is limited due to the different experimental protocols employed. Furthermore, the manufacturer’s data sheets typically reference bulk samples or samples printed at a 100% infill density, whereas the present study utilized an 80% of infill density.

Subsequent to the analysis of the mechanical properties of commercially available unmodified polymers, the analysis of the modified nanocomposite ferromagnetic filaments was conducted. The subsequent stress–strain curves (shown in Figure 2a) illustrate and interpret the mechanical response of these materials as the weight percentage of nanoparticles increases, along with the potential mechanisms underlying this behavior.

As extensively discussed in the literature, metal oxides have been utilized as fillers to modify polylactic acid (PLA) nanocomposites, broadening their application scope [32,33]. However, incorporating nanoparticles into polymeric matrices presents challenges due to their difficult dispersion within the polymer [34]. To explore the effect of dispersion on PLA properties, SiO_2_ systems have been superficially modified, and a coupling agent has been employed to enhance the dispersion of SiO_2_ nanoparticles [35]. These modifiers are commonly used to alter the surface charge of metal oxide nanoparticles, improving their stability in solvents and polar matrices.

Moreover, the particle size of the filler significantly influences the crystallization process; micro-sized fillers tend to enhance the crystallization rate and growth of PLA, whereas a high concentration of nanometric particles may inhibit initial crystal growth [36,37]. This retardation is likely due to the reduced mobility of PLA molecules caused by the presence of nanoparticles [38,39].

Previous studies have demonstrated that particle size and surface area significantly influence the interaction between polymer chains and nanoparticles [40,41]. Young’s modulus measurements indicate that nanocomposites reinforced with spherical nanoparticles exhibit the highest rigidity. Furthermore, the rate of crystal formation plays a critical role in determining the mechanical properties of nanocomposites composed of nanoparticles with a small surface area [42,43]. Conversely, when nanocomposites are reinforced with high-surface-area nanoparticles, the number of crystals formed markedly affects their mechanical properties. This phenomenon occurs because polymer chains are arranged differently around the nucleation centers [44].

Based on the preceding analysis, we can interpret the observed changes in the mechanical properties of the fabricated nanocomposite materials using the mechanical characterization techniques described. Initially, the reduction in the mechanical properties of EF. PLA with the incorporation of 10 wt% magnetite nanoparticles can be attributed to the analyzed mechanisms. The relatively uneven dispersion of nanoparticles, combined with their spherical shape and lack of surface modification, hinders their ability to form effective bonds with the polylactic acid chains. As a result, these nanoparticles act as centers of local stress concentration under mechanical loading, leading to diminished mechanical performance.

Increasing the concentration of nanoparticles to 20 wt% exacerbates these effects, further degrading the mechanical properties of the printed structures. Additionally, changes in the melting temperature of the nanocomposite polymers constitute another mechanism influencing mechanical performance, which is related to the increased concentration of the dispersed phase of nanoparticles. Variations in the concentration of nanoparticles affect the melting temperature and, consequently, the flow of the molten polymer during 3D printing. As reported in the literature, reduced polymer flow impedes the diffusion of polymer chains between deposited layers, resulting in a diminished mechanical response of the final structures [45,46,47]. To illustrate these findings, the mechanical behavior (stress–strain curves) of the following families of materials—(a) EF.PLA, (b) PR.PLA, and (c) T.PLA—is presented in Appendix A as a function of increasing magnetite concentration, depicting the relationship between nanoparticle concentration and the elastic modulus of the printed structures. The Young’s modulus of the fabricated filaments is also presented concerning the concentration of magnetic nanoparticles in Figure 3a. It is clearly shown in the bar diagrams of Figure 3a that, in the case of EF. PLA, Young’s modulus decreases gradually as the concentration of nanoparticles increases. On the contrary, we observe a different behavior with T. PLA. Initially, increasing the nanoparticle concentration from 10 to 20 wt% results in an increase in the modulus of elasticity. This behavior can be explained by the mechanism described by A. Sanida et al. [48], where the increase in storage modulus values with increasing nanoparticle content is attributed to the restriction of molecular motion within the amorphous portions of the polymer chains.

These chains are confined within the crystalline regions or at the crystalline/amorphous interface due to the presence of various types of defects. These defects, either in the crystal structure or at the interface, limit the mobility of the polymer chains, contributing to the observed rise in modulus.

However, when the concentration of magnetite is increased to 20 wt%, the modulus of elasticity decreases dramatically. This decrease can be attributed to excessive nanoparticle loading, which likely disrupts the polymer structure, resulting in agglomeration and poorer dispersion. Consequently, this leads to an overall reduction in the mechanical performance of the composite material.

This phenomenon can be attributed to an increase in the number of defects within the polymer chains, likely caused by printing flaws that enhance the overall fragility of the structure. In the case of Pr. PLA, this behavior was not observed; instead, there was almost no change in the modulus of elasticity with increasing nanoparticle concentration, even at a magnetite concentration of 20 wt%.

In Figure 3b, which depicts the ultimate tensile stress and in Figure 4a which illustrates the stress at the yield point as a function of nanoparticle concentration, it was observed that both values decreased as nanoparticle concentration increased. However, for EF. PLA, the increase in nanoparticles did not result in a significant decrease in these values; in fact, they remained relatively stable within the same limits. To further assess whether the increase in concentration significantly impacts mechanical properties, it is essential to examine how the maximum hardening before fracture is affected. For this purpose, the bar graph of Figure 3b was generated to correlate nanoparticle concentration with the pre-fracture strength of the structures.

In Figure 4a, the bar diagram illustrates the yield strength (σ_y_) as a function of magnetite concentration, showing that as the concentration increases from 0 to 20 wt%, the σ_y_ is reduced for all types of composite filaments. Notably, at 20 wt% magnetite concentration, the yield strength for all three filament types reaches just above 20 MPa. Among these, T.PLA exhibits the lowest yield strength value when considering the non-magnetic filament. This trend highlights the impact of magnetite concentration on the mechanical properties of the composite filaments, emphasizing the trade-offs in yield strength associated with increased magnetic nanoparticle content.

In this context, Figure 4b shows that the maximum elongation before fracture follows a similar trend—i.e., as the concentration of magnetite nanoparticles increases, the resulting elongation decreases. T. PLA, which exhibits the highest impact resistance among all the materials tested, as reported by the manufacturer, shows a slightly different behavior. One possible explanation is that T. PLA’s high impact resistance allows greater freedom for polymer chains to move relative to one another, enabling them to absorb tensile energy plastically when tensile loads are applied and thus resulting in greater strains before fracture [49,50,51].

To evaluate the mechanical performance of the filaments, stress–strain experiments were conducted, and the results are summarized in Appendix A. The table presents the elastic modulus (E), ultimate tensile strength (σ_UTS_), yield strength (σ_y_), and elongation at break (ε) for three types of neat PLA materials (EasyFill PLA, Tough PLA, and Premium PLA). Additionally, these mechanical properties are reported for PLA filaments mixed with 10 and 20 wt% magnetic nanoparticles. For a comprehensive comparison, the percentage differences in these properties between the mixed PLA and the corresponding neat PLA are also provided.

The percentage differences (ΔE, Δσ_UTS_, Δσ_y_, and Δε) for the mixed PLA samples (10 and 20 wt% of magnetic nanoparticles) are calculated using the following formula:$$\Delta \left(\%\right)=\frac{Value\;for\;mixed\;PLA-Value\;for\;neat\;PLA}{Value\;for\;neat\;PLA}\;\times \;100\;$$

These results indicate that the incorporation of magnetic nanoparticles into PLA filaments significantly alters their mechanical properties. Specifically, the elastic modulus, ultimate tensile strength, yield strength, and elongation at break exhibited noticeable reductions with the increase in nanoparticle concentration, particularly at higher values (in our case 20 wt%). This is clearly shown in Appendix A columns presenting the percentage differences (ΔE, Δσ_UTS_, Δσ_y_, and Δε) for the mixed PLA samples (10 and 20 wt% of magnetic nanoparticles). This trend aligns with findings reported by Constant-Mandiola et al. [52], which illustrated that mechanical properties of PLA composites are significantly influenced by both the filler content and also by its nature. In their work, they noted that a 7% filler of magnetite nanocomposite nanoparticles can lead to a 28% decrease in Young’s modulus, a result comparable with the percentage differences of Young’s modulus of T. PLA composite filaments. In our work, we utilized EasyFil PLA to produce filaments using an extruder set at a temperature of 165 °C. In contrast, Tough PLA and Premium PLA required slightly higher extrusion temperatures due to their distinct formulations. Notably, filaments incorporating Fe_3_O_4_ nanoparticles exhibited a need for an additional 5–10 °C above standard extrusion settings, attributable to the increased viscosity of the composite material. When comparing these processing conditions to the study conducted by Constant-Mandiola et al., which employed extrusion temperatures ranging from 160 to 180 °C, it becomes evident that the thermal conditions are critical for optimizing the mechanical properties of the resulting nanocomposites. The elevated temperatures necessary for processing our Fe_3_O_4_ nanoparticle composites may raise concerns regarding potential polymer degradation, particularly as reported in [52]. Specifically, prolonged exposure to such temperatures can lead to the breakdown of polymer chains, resulting in a reduction in molecular weight and crystallinity, which may adversely affect mechanical performance. Furthermore, in a recent study [53], the incorporation of magnetic nanoparticles into PLA was explored at varying concentrations of 10, 15, and 20 wt%. Their findings revealed that the addition of 10 wt% Fe_3_O_4_ nanoparticles enhanced the mechanical properties of the PLA blends, resulting in a 16% increase in ultimate tensile strength (σ_UTS_), achieving a value of 35.89 MPa. Notably, this value is very close to the σ_UTS_ of our sample, which achieved 33 MPa for the Pr. PLA filament fabricated with 10 wt% Fe_3_O_4_ nanoparticles. However, the study also indicated that further increases in Fe_3_O_4_ concentration to 15 wt% and 20 wt% led to a decline in mechanical performance, with σ_UTS_ decreasing to 29.8 MPa and 27.7 MPa, respectively. These results are consistent with our findings, where the ultimate tensile strength (σ_UTS_) for PLA samples mixed with 20 wt% magnetite reached values ranging from 25 to 30 MPa. This reduction was primarily ascribed to nanoparticle agglomeration and the resulting elevated melt viscosity, which hindered effective dispersion and bonding within the polymer matrix.

### 3.3. Magnetic Hyperthermia Evaluation

Magnetic hyperthermia measurements were performed on all material families. In the temperature–time diagrams presented in Figure 5a–c, we can observe how the temperature rises when an alternating magnetic field is applied, for the magnetic, Figure 5d,f,g and the reference non-magnetic, Figure 5e samples.

The temperature increased in all magnetic scaffolds under the application of an alternating (AC) magnetic field of 20 mT with a frequency of 400 kHz. On the other hand, the non-magnetic samples (agarose only and Neat PLA) kept their initial temperature under the application of the AC field, indicating that the thermal efficiency of the magnetic samples is due to their intrinsic magnetic properties.

Initially, for the samples of the EF.PLA family, we observe that there is no temperature change associated with the applied magnetic field, as there is no interaction between the magnetic field and materials in which magnetite is not dispersed. With the addition of 10 wt% magnetite (EF. PLA10%), we observe a rapid increase in temperature to the limits of magnetic hyperthermia in a very short period of time (about 25 s).

With an increase in magnetite content in the EF. PLA20% sample, we observe that the structures surpass the magnetic hyperthermia temperature limit after applying an alternating magnetic field (AC) of 20 mT at a frequency of 400 kHz, achieving this in less than 20 s. This finding outlines the efficiency of magnetic hyperthermia and its enhancement during the application of the alternating magnetic field is directly proportional to the concentration of nanoparticles. Comparing this with the EF. PLA10% sample, it is evident that increasing the concentration of magnetic nanoparticles enhances the heating efficiency of the scaffold.

Regarding the T. PLA and Pr. PLA composites, we observe similar behavior across these materials and their nanocomposite structures. In the case of T. PLA 10%, the structures reach the threshold of magnetic hyperthermia in approximately 50 s, whereas doubling the percentage of magnetite nanoparticles reduces this time by half. The behavior of Pr. PLA and its associated structures is comparable to that observed in T. PLA. For Pr. PLA 10%, the scaffold approaches temperatures of 41–45 °C in roughly 50 s, and similarly, doubling the amount of nanoparticles in this case also decreases the time required for the structures to enter the anticipated hyperthermia window.

This phenomenon occurs in a very short amount of time and can be attributed to the high nanoparticle concentration. We also observed that the cooling time is comparable in all three instances. From this analysis, we can conclude that the two proportions chosen to build the smart structures approach two quite extreme cases (especially the one with the 20 wt% of magnetite nanoparticles is quite large and, as we have seen, causes problems in the printing processes and has a significant impact on the mechanical properties); therefore, nanoparticle percentages lower than 10 wt% of magnetite nanoparticles could give us better control over the magnetic hyperthermia process, while allowing us to remain within the desired range.

In our previous work [20], the distinction between the heating evaluation of magnetic nanoparticles (MNPs) and magnetic scaffolds was examined and confirmed. The specific loss power (SLP) and the specific absorption rate (SAR) were discussed in detail, emphasizing the importance of using SAR for assessing the heating performance of magnetic scaffolds. Since heat is dissipated into the scaffold’s surrounding medium (agarose in our case, or tissue in in vivo applications), SAR, calculated using the following Equation (1), provides a reliable measure of the scaffold’s heating efficiency.(1)$$SAR=c\left(\frac{\Delta T}{\Delta t}\right)$$
where ΔT Δt^−1^ (°C s^−1^) is the initial heating rate, and c (J/kg·°C) is the specific heat capacity of the agarose gel phantom, set to 4184 J/kg·°C.

Figure 6 presents the SAR values of the composite scaffolds, measured under these conditions. A clear trend is observed: for all PLA-based scaffolds, SAR increases proportionally with magnetite content. Specifically, doubling the magnetite concentration from 10 to 20 wt% leads to an almost twofold increase in SAR. This trend is consistent with our previous findings [20], reinforcing that the heating efficiency of composite scaffolds is governed by the concentration and distribution of magnetic nanoparticles within the polymer matrix.

Moreover, according to Figure 6, among the different PLA filaments used, scaffolds fabricated with T. PLA exhibit the highest SAR values compared to those made by EF. PLA and Pr. PLA. This enhanced thermal efficiency can be attributed to a more uniform dispersion of magnetite NPs within the T. PLA matrix, optimizing the conversion of magnetic energy into heat. Additionally, the superior mechanical properties of T. PLA further support its suitability for magnetic hyperthermia applications, making it the most promising material among the tested composites.

These results confirm that the selection of the polymer matrix plays a crucial role in determining the heating performance of 3D-printed magnetic scaffolds. By optimizing both the polymer type and magnetite concentration, it is possible to fine-tune the thermal response of scaffolds for biomedical applications such as targeted hyperthermia therapy and tissue engineering.

Our previous studies [15,19,20] have established the significant impact of magnetite nanoparticle content on the magnetic properties and heating efficiency of composite scaffolds, particularly in the context of magnetic hyperthermia applications. These findings highlighted that by increasing the magnetite content in non-magnetic filaments like PLA, the magnetic properties and the heating efficiency of the scaffolds can be enhanced, making them suitable for thermal therapies. Additionally, the magnetic profile of scaffolds printed with commercially available magnetic filaments can be adjusted by varying the percentage of magnetite NPs, which also affects their heating performance.

In this work, we go further by examining the role of thermoplastic filament choice in the final magnetic hyperthermia application. While our previous studies focused mainly on the influence of magnetite nanoparticle content, this study extends that understanding by exploring how different thermoplastic materials, such as EasyFil PLA, Premium PLA, and Tough PLA, influence the performance of the scaffolds in magnetic hyperthermia. The thermoplastic matrix not only affects the mechanical properties and printability of the scaffolds but also plays a key role in their thermal response. Our results show that by carefully selecting the right thermoplastic material, we can balance both mechanical integrity and thermal efficiency, making the Tough PLA composite the most promising material for magnetic hyperthermia applications, as it provides optimal heating performance without compromising scaffold strength. The SAR values of the composite scaffolds ranged from 2 to 7.5 W/g, demonstrating the variability in thermal response depending on the material composition and magnetite concentration.

## 4. Conclusions

In this study, synthesized nanocomposite ferromagnetic filaments were fabricated to demonstrate their application in constructing bone tissue scaffolds using 3D printing techniques. Three modified biocompatible polymers derived from polylactic acid (EasyFil PLA, Premium PLA, Tough PLA) were used, along with two structure types and varying magnetite nanoparticle (MNP) concentrations (10 and 20 wt%). The incorporation of MNPs significantly impacted both the mechanical properties and thermal performance of the printed scaffolds, particularly with respect to their heating efficiency action.

The mechanical properties of the filaments exhibited a decreasing trend with the incorporation of MNPs. While the addition of 10 wt% MNPs resulted in comparable tensile strength to the neat polymer, further increasing the MNP concentration to 20 wt% led to a reduction in ultimate tensile strength and toughness. This decline is attributed to nanoparticle agglomeration and increased viscosity during extrusion, which negatively impact the structural integrity of the filaments. These findings emphasize the necessity of optimizing MNP loading to ensure a balance between magnetic functionality and mechanical stability, particularly for tissue engineering applications.

Thermally, the scaffolds exhibited increased efficiency in magnetic hyperthermia as the MNP concentration rose. Scaffolds containing 20 wt% MNPs reached the hyperthermia temperature range in less than 20 s under an alternating magnetic field, outperforming those with lower concentrations. The SAR results further reinforce this trend, showing that for all PLA-based scaffolds, SAR values nearly double when the MNP content is increased from 10 to 20 wt%. Among the different PLA matrices tested, Tough PLA scaffolds incorporating 10 and 20 wt% magnetite nanoparticles exhibited the highest SAR values, likely due to the improved dispersion of MNPs within their structure. This makes Tough PLA composites the most promising material for hyperthermia applications, as they offer an optimal balance of thermal efficiency and mechanical integrity compared to the other two PLA variants.

Additionally, the study explored the potential of 4D printing by designing scaffolds with properties that could evolve. This adaptability is particularly promising for applications such as bone tissue regeneration, where gradual changes in scaffold properties may align with biological processes. The findings of this study contribute to the optimization of 3D-printed magnetic scaffolds for biomedical applications, particularly in tissue engineering and magnetic hyperthermia-based treatments.

## Figures and Tables

**Figure 1 micromachines-16-00328-f001:**
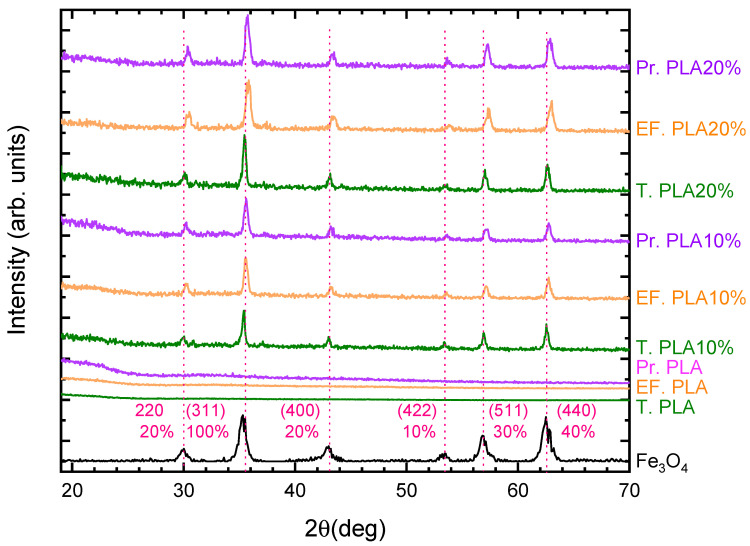
Comparative X-ray diffraction pattern for all PLA materials (T. PLA, EF. PLA and Pr. PLA) with and without magnetite MNPs (10 and 20 wt%). The percentages below the h, k, l values correspond to the relative intensity of each diffraction peak, scaled to the strongest peak, which is assigned an intensity of 100.

**Figure 2 micromachines-16-00328-f002:**
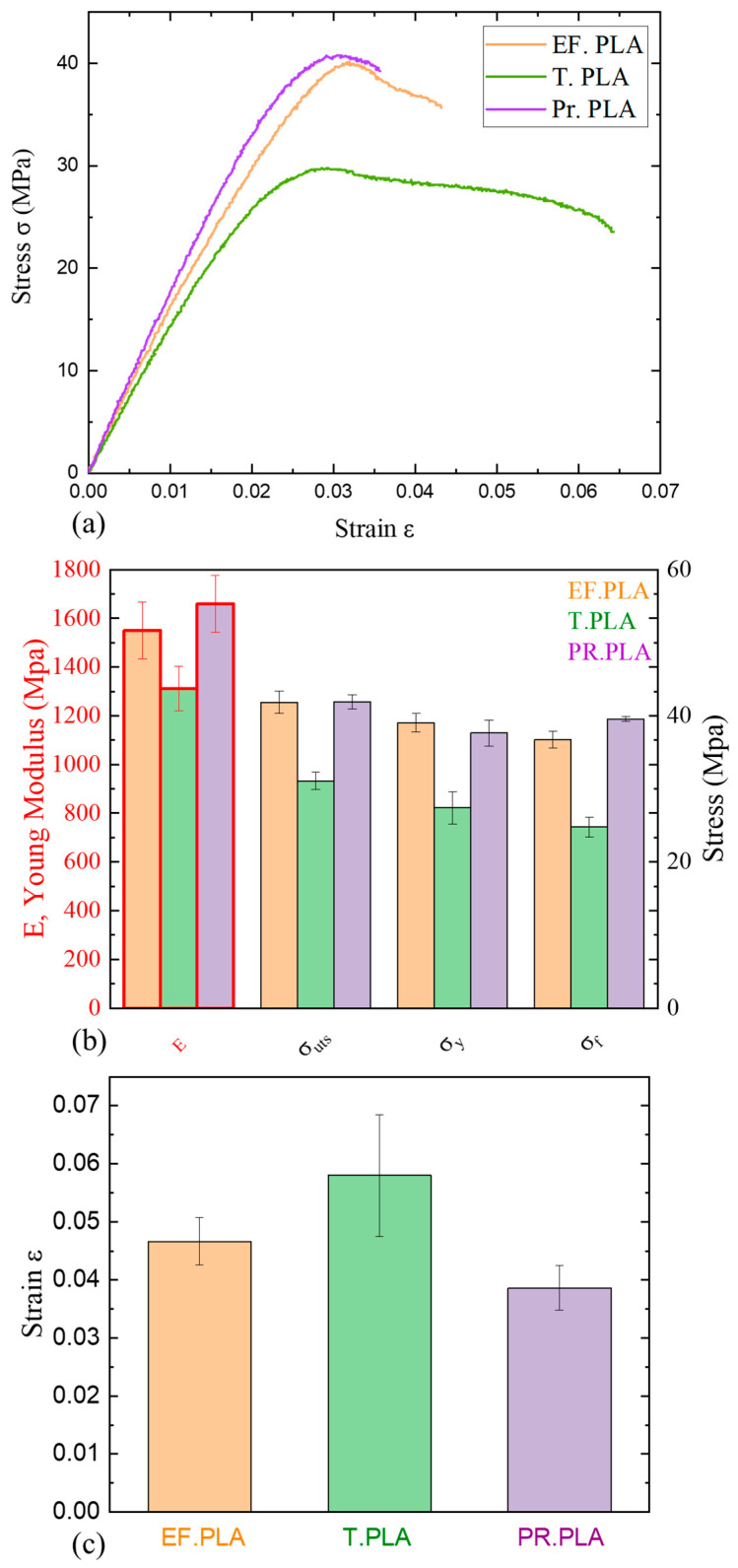
(**a**) Comparative stress–strain curves of the 3D-printed neat materials for the three PLA materials (T. PLA, EF. PLA and Pr. PLA), with (**b**) their respective mechanical properties. (**c**) Comparative column bar diagrams of elongation of the above PLA materials.

**Figure 3 micromachines-16-00328-f003:**
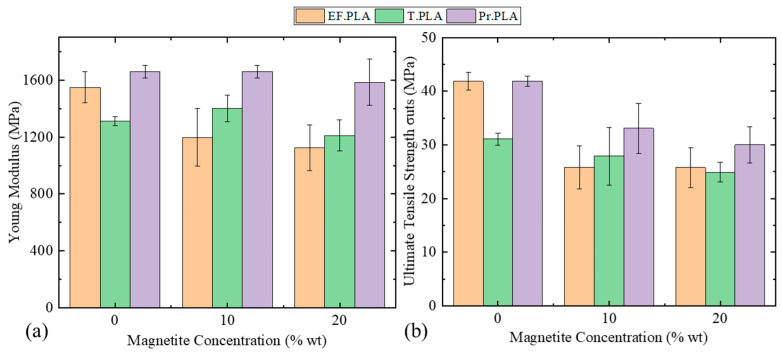
Influence of magnetite concentration on the modulus of elasticity and ultimate tensile strength. The effects of varying magnetite concentrations on (**a**) the modulus of elasticity and (**b**) the ultimate tensile strength are illustrated.

**Figure 4 micromachines-16-00328-f004:**
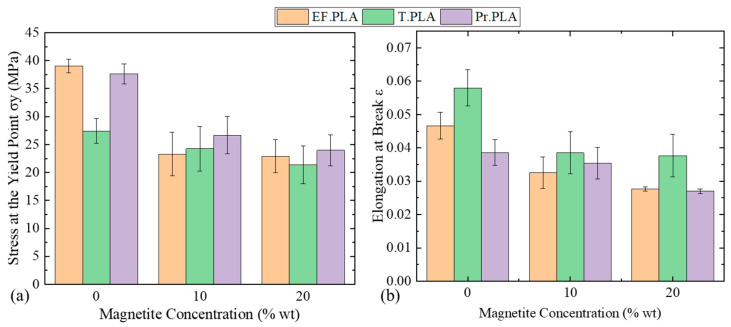
Influence of magnetite concentration on stress at yield point and elongation at break. The effects of varying magnetite concentrations on (**a**) stress at the yield point and (**b**) elongation at break are presented.

**Figure 5 micromachines-16-00328-f005:**
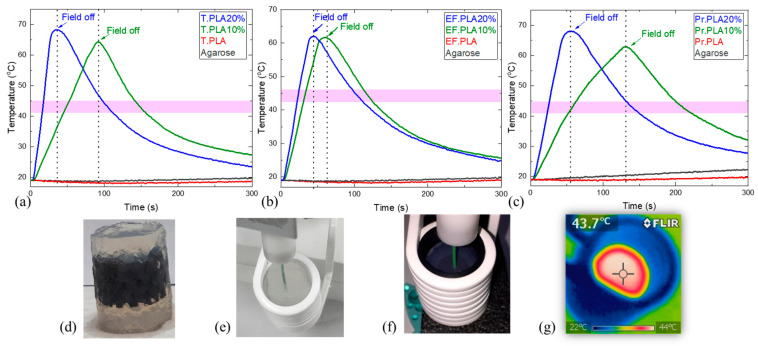
Temperature–time profiles of 3D-printed magnetic scaffolds under an alternating magnetic field (20 mT, 400 kHz). The graphs illustrate the hyperthermia behavior of (**a**) Tough PLA (T.PLA), (**b**) EasyFil PLA (EF.PLA), and (**c**) Premium PLA (Pr.PLA). Each material is depicted with its magnetic samples containing 10 wt% (green) and 20 wt% (blue) magnetite nanoparticles. The vertical dotted lines to the time axis represent the points when the AC magnetic field is turned off. A colored text field labeled “Field off” has also been added, with arrows pointing to the moment when the field is turned off, providing a clearer visual representation. Reference samples, including Neat PLA (red) and agarose only (gray), are also included, showing no significant temperature change under the same conditions. Additionally, (**d**–**f**) are photos corresponding to the magnetic scaffold placed in agarose gel, agarose positioned in the center of the 8th turn coil, and the magnetic scaffold in agarose placed in the center of the induction heating coil, respectively. The optical fiber shown in green in photos (**e**,**f**) is positioned at the center of the samples to record the temperature. The infrared photo shown in (**g**) provides information about heat diffusion around the scaffold sample, indicating that the temperature of the scaffold has reached 43.7 °C.

**Figure 6 micromachines-16-00328-f006:**
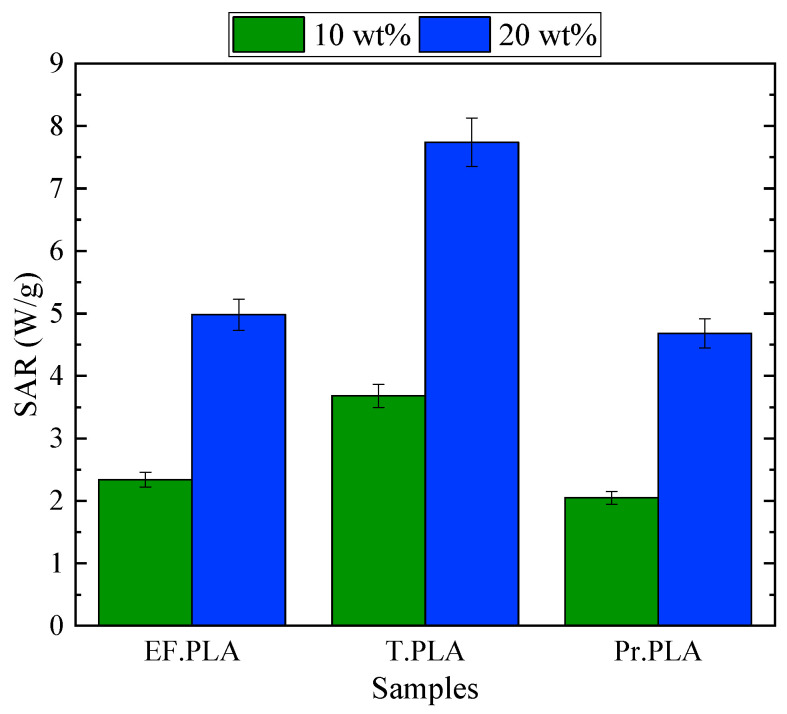
SAR values for the magnetite/thermoplastic composite scaffolds 3D printed with filaments containing varying magnetic content (10 and 20 wt% magnetite NPs in PLA, represented by green and blue colors, respectively). The thermoplastic PLA used for each filament type includes Premium PLA (Pr. PLA), EasyFil PLA (EF. PLA), and Tough PLA (T. PLA). The evaluated SAR values correspond to the magnetic hyperthermia response of the magnetic scaffolds under an external field of 20 mT at a frequency of 400 kHz.

**Table 1 micromachines-16-00328-t001:** Summary of the 3D-printed scaffolds and dog bone samples.

Sample	Filament Type	MNP Concentration (wt% MNPs)	Scaffold Dimensions (mm)	Dog Bone Dimensions (mm)
Pr. PLA	Pr. PLA	0	16 (H) × 19 (D)	
Pr. PLA10%		10		63.5 (L) × 9.53 (G)
Pr. PLA20%		20		
EF. PLA	EF. PLA	0	16 (H) × 19 (D)	
EF. PLA10%		10		63.5 (L) × 9.53 (G)
EF. PLA20%		20		
T. PLA	T. PLA	0	16 (H) × 19 (D)	
T. PLA10%		10		63.5 (L) × 9.53 (G)
T. PLA20%		20		

## Data Availability

The original contributions presented in the work are included in the article: further inquiries can be directed at the corresponding author.

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
