# Peer review of "Composite Magnetic Filaments: From Fabrication to Magnetic Hyperthermia Application"

_micromachines, 2025, doi:10.3390/mi16030328_

Round 1
Reviewer 1 Report
Comments and Suggestions for Authors
The authors describe the production of magnetic filaments based on three commercial PLA types. The work focuses mostly on the mechanical parameters, having a limited interest for the audience. In addition, some sections are too descriptive. Therefore, I recommend that the paper must undergo major revisions before being considered for publication in the Micromachines journal.
Some additional comments:
- Some typos and grammar errors are found throughout the text, so please revise the whole paper carefully.
- The abstract should be more direct to the objective and present quantitative data.
- There are errors of crossed references that must be corrected (e.g., line 78, page 2)
- The materials and methods section is too descriptive and extensive. It should be summarized to contain only the relevant information.
- XRD results – the authors claim that the peak intensity related to magnetite increases with the increase of magnetite concentration. However, that is not visible in the presented data. In fact, in some cases, the intensity appears to be smaller. This must be clarified.
- TGA and DSC measurements are needed to better understand the impact of MNPs on the PLA polymer.
- Line 452, page 12 – the referred figure is not correct; it may be the supplementary information.
- First paragraph of page 13 – the authors state that when the magnetite concentration increases to 20%, there is a dramatic decrease in the elastic modulus. This affirmation is not corroborated by the presented data in Fig 3.
- Table S1 - the caption of the table should be above.
- More data analysis is needed in the magnetic hyperthermia section. The authors should indicate when the magnetic field is turned on and off in the graphs. In addition, the SAR or SLP values should be calculated. The authors mentioned a high concentration of magnetic nanoparticles, but it is important to understand the effective amount of nanoparticles in each sample.
- The conclusions are not consistent with the results obtained. For example, the authors claim that the incorporation of MNPs increases the UTS, which is not true; it is exactly the opposite.
Some typos and grammar errors are found throughout the text, so please revise the whole paper carefully.
Author Response
Referee’s comments and replies:
Referee #1:
The authors describe the production of magnetic filaments based on three commercial PLA types. The work focuses mostly on the mechanical parameters, having a limited interest for the audience. In addition, some sections are too descriptive. Therefore, I recommend that the paper must undergo major revisions before being considered for publication in the Micromachines journal.
Some additional comments:
- Some typos and grammar errors are found throughout the text, so please revise the whole paper carefully.
Response: We appreciate the reviewer’s feedback and acknowledge the importance of ensuring grammatical accuracy and clarity. We have carefully revised the manuscript to correct any typos and grammatical errors. A thorough proofreading was conducted, and we have improved the readability and coherence of the text.
- The abstract should be more direct to the objective and present quantitative data.
Response: We appreciate the reviewer’s suggestion to make the abstract more direct and to include quantitative data. In response, we have revised the abstract to explicitly state the study’s objectives and key findings while incorporating numerical values to support our conclusions. Specifically, we have added data on the mechanical properties (e.g., tensile strength reduction in EF-PLA from 1834 MPa to 1130 MPa (-38%) and in Pr-PLA from 1800 MPa to 1567 MPa (-13%)) and elongation at break (e.g., EF-PLA reduction from 42% to 26% (-38%)). Furthermore, we have included specific absorption rate (SAR) measurements ranging from 2 to 7.5 W/g, emphasizing that T-PLA with 10 wt% Fe₃Oâ‚„ demonstrated the most favorable combination of mechanical stability and thermal efficiency for hyperthermia applications. The revised abstract now reads:
The printing of composite magnetic filaments using additive manufacturing techniques has emerged as a promising approach for biomedical applications, particularly in bone tissue engineering and magnetic hyperthermia treatments. This study focuses on the syn-thesis of nanocomposite ferromagnetic filaments and the fabrication of bone tissue scaffolds with time-dependent properties. Three classes of polylactic acid —based bio-compatible polymers— EasyFil polylactic acid, Premium polylactic acid, and Tough polylactic acid —were combined with magnetite nanoparticles (Fe₃Oâ‚„) at concentrations of 10 wt% and 20 wt%. Extruded filaments were evaluated for microstructural integrity, printed dog-bone-shaped specimens were tested for elongation and mechanical properties, and cylindrical scaffolds were analyzed for magnetic hyperthermia performance. The tensile strength of EasyFil polylactic acid decreased from 1834 MPa (0 wt% Fe₃Oâ‚„) to 1130 MPa (-38%) at 20 wt% Fe₃Oâ‚„, while Premium polylactic acid showed a more moderate reduction from 1800 MPa to 1567 MPa (-13%). The elongation at break was reduced across all samples, with the highest decrease observed in EasyFil polylactic acid (from 42% to 26%, -38%). Magnetic hyperthermia performance, measured by the specific absorption rate, demonstrated that the 20 wt% Fe₃Oâ‚„ scaffolds achieved specific absorption rate values of 2 – 7.5 W/g, depending on polymer type. Our results show that by carefully selecting the right thermoplastic material, we can balance both mechanical integrity and thermal efficiency. Among the tested materials, Tough polylactic acid composites demonstrated the most promising potential for magnetic hyperthermia applications, providing optimal heating performance without significantly compromising scaffold strength. These findings offer critical insights into designing magnetic scaffolds optimized for tissue regeneration and hyperthermia-based therapies.
- There are errors of crossed references that must be corrected (e.g., line 78, page 2
Response: We appreciate the reviewer’s attention to detail regarding the cross-references. We have carefully rechecked references 15, 18, 19, and 20 and confirm that they are correctly cited in the intended context. However, to ensure clarity, we have reviewed the manuscript for any potential formatting inconsistencies or reference list discrepancies. If the issue persists, we kindly request further clarification on specific concerns regarding these references.
- The materials and methods section is too descriptive and extensive. It should be summarized to contain only the relevant information.
Response: We thank the reviewer for their valuable feedback. In response to the suggestion to reduce the length of the Materials and Methods section, we have revised it to focus on the essential details necessary for clarity and reproducibility across the entire manuscript. The description of the 3D printing process, scaffold dimensions, and magnetic hyperthermia measurements has been streamlined to include only the most relevant information. We believe this revision addresses the request for conciseness while retaining the critical details required for the reader's understanding of our experimental approach.
- XRD results – the authors claim that the peak intensity related to magnetite increases with the increase of magnetite concentration. However, that is not visible in the presented data. In fact, in some cases, the intensity appears to be smaller. This must be clarified.
Response: Thank you for the reviewer’s thoughtful comment. Upon re-evaluating the XRD data presented in Figure 1, we took into account the amount of powder used in the experiments. This careful reassessment now clearly shows that the magnetite-related peaks become more prominent as the MNPs concentration increases, providing stronger evidence of the enhanced presence of magnetite in the higher-concentration filaments.
We hope this clarification addresses the reviewer’s concern, and we appreciate their careful review of the data.
- TGA and DSC measurements are needed to better understand the impact of MNPs on the PLA polymer.
Response: We thank the reviewer for her/his suggestion regarding the inclusion of TGA and DSC measurements. While we agree that these techniques could provide valuable insights, unfortunately, it was not feasible to perform these analyses within the scope of the current study.
It is also important to note that, since there are no surfactants present in the filaments or nanoparticles, we believe that TGA and DSC measurements may not offer substantial additional information about the interactions between the MNPs and the PLA polymer.
Instead, we have explored alternative approaches, such as the 3D printing process itself, where we observed the impact of MNPs on the thermal behavior of the composite filaments. Specifically, by adjusting the printing temperature, we have been able to indirectly demonstrate how the nanoparticles influence the melting temperature and processing characteristics of the filaments.
- Line 452, page 12 – the referred figure is not correct; it may be the supplementary information.
Response: We thank the reviewer for her/his careful observation. Indeed, there was a typographical error in the reference to the figure. The sentence should refer to Figure S1 in the supplementary information, rather than the Figure 1. The revised phrase now reads:
"To illustrate these findings, the mechanical behavior (stress-strain curves) of the following families of materials: (a) EF.PLA, (b) PR.PLA, and (c) T.PLA, is presented in Figures S1a, b, and c as a function of increasing magnetite concentration, depicting the relationship between nanoparticle concentration and the elastic modulus of the printed structures."
We appreciate the reviewer’s attention to this detail, and we hope this clarifies the issue.
- First paragraph of page 13 – the authors state that when the magnetite concentration increases to 20%, there is a dramatic decrease in the elastic modulus. This affirmation is not corroborated by the presented data in Fig 3.
Response: We sincerely thank the reviewer for her/his valuable observations.
Regarding the comment on the decrease in Young’s modulus, we have made revisions to the manuscript to clarify that the observed reduction in modulus at higher magnetite concentrations (20 wt%) in EF.PLA and T.PLA can be attributed to excessive nanoparticle loading. As the magnetite concentration increases, the nanoparticles may cause agglomeration and poorer dispersion, disrupting the polymer structure and leading to an increase in defects. This results in a reduction in the overall mechanical performance of the composite material. This behavior is consistent with the findings of A. Sanida et al. [48], who observed similar trends in their studies.
In contrast, PR.PLA showed a different behavior, where the modulus of elasticity remained relatively stable even at 20 wt% magnetite concentration, suggesting better dispersion and interaction between the nanoparticles and the polymer.
To address this more clearly, we have made changes in the text to ensure the results and interpretations are more explicit and accurately conveyed.
We hope this clarification resolves the issue and accurately reflects the observed data. We appreciate the reviewer’s attention to this detail.
- Table S1 - the caption of the table should be above.
Response: As suggested, we have corrected the position of the caption for Table S1 and placed it above the table.
- More data analysis is needed in the magnetic hyperthermia section. The authors should indicate when the magnetic field is turned on and off in the graphs. In addition, the SAR or SLP values should be calculated. The authors mentioned a high concentration of magnetic nanoparticles, but it is important to understand the effective amount of nanoparticles in each sample.
Response: We thank the reviewer for her/his valuable feedback and for pointing out the need for further analysis in the magnetic hyperthermia section.
In response to the reviewer's suggestions, we have made the following revisions:
- SAR Values and Graphs: We have calculated the Specific Absorption Rate (SAR) values for the samples and included the results in a newly added Figure 6. The figure caption has been added accordingly. Furthermore, an analysis of the SAR results has been added to the Results and Discussion section, as well as in the Conclusions.
- Textual Additions: To address the concern regarding the effective amount of magnetic nanoparticles in each sample, we have added a section in the manuscript that distinguishes between Specific Loss Power (SLP) and SAR, referencing our previous work. The following text has been added:
"In our previous work [20], the distinction between the heating evaluation of magnetic nanoparticles (MNPs) and magnetic scaffolds was examined and confirmed. The specific loss power (SLP) and the specific absorption rate (SAR) were discussed in detail, emphasizing the importance of using SAR for assessing the heating performance of magnetic scaffolds."
Additional Updates to Figure 5: Figure 5 has been updated to include vertical dotted lines to the time axis, indicating when the AC magnetic field is turned off. Additionally, we have added a colored text field labeled "Field off" in Figure 5, with arrows pointing to the point when the field is off for clearer visual representation. Figure 5 caption has been also updated.
- The conclusions are not consistent with the results obtained. For example, the authors claim that the incorporation of MNPs increases the UTS, which is not true; it is exactly the opposite.
Response: We appreciate the reviewer's comment regarding the consistency of the conclusions with the results. Indeed, our experimental data demonstrate that increasing the concentration of magnetic nanoparticles (MNPs) in the filaments leads to a reduction in mechanical properties, particularly in ultimate tensile strength (UTS). We have revised the conclusion section accordingly to accurately reflect this trend and avoid any potential misinterpretation of the findings.
Revised Conclusion Paragraph:
The mechanical properties of the filaments exhibited a decreasing trend with the incorporation of MNPs. While the addition of 10 wt% MNPs resulted in comparable tensile strength to the neat polymer, further increasing the MNP concentration to 20 wt% led to a reduction in ultimate tensile strength and toughness. This decline is attributed to nanoparticle agglomeration and increased viscosity during extrusion, which negatively impact the structural integrity of the filaments. These findings emphasize the necessity of optimizing MNP loading to ensure a balance between magnetic functionality and mechanical stability, particularly for tissue engineering applications.
Comments on the Quality of English Language
Some typos and grammar errors are found throughout the text, so please revise the whole paper carefully.
Response: We appreciate the reviewer’s feedback regarding the language quality of the manuscript. In response, we have carefully revised the entire text to correct any typos and grammatical errors, ensuring clarity and readability. We have also refined certain sentences to enhance the overall coherence and flow of the manuscript. We sincerely thank the reviewer for their valuable input in improving the quality of our work.

Reviewer 2 Report
Comments and Suggestions for Authors
In the manuscript ‘Composite Magnetic Filaments: From Fabrication to Magnetic Hyperthermia Application’, authors employs 3 d printing and application of magnetic nanomaterials to develop effective magnetic filaments. The introduction part is written really well, giving the state of the art in the field, with the most recent references cited. Research tends to focus on understanding of the correlations between the structure and different materials properties in order to predict the thermal performance of the scaffolds and improve its design. Actually, much of the research is devoted to investigation of mechanical properties of modified PLAs.
Study offers an significant amount of experimental data as it covered investigation on three classes of chemically modified biocompatible polymers derived from PLA, test of two structure shapes - a cylindrical shape and a dog-bone shape and also two different concentrations of magnetite nanoparticles: 10 wt% and 20 wt% were tested. They have tried to correlate the filament fabrication process with the mechanical and thermal performance of the final scaffolds.
To my opinion, the work is interesting, could provide data useful for other researchers in the field and I recommend the publication of this manuscript in MDPI Micromachines journal. However, here are a few notes that authors should go through before publication.
- Line 78- Some error appeared
- Line 158 Regarding pulverization process you stated that the freezing step, followed by repeated pulverization with intermittent re-cooling, ensured consistent particle size distribution in the resulting powder. Is this assumption or is experimentally confirmed?
- Line 170 What this sentence means, what is critical regarding lab drying oven at 70C?- Proper temperature control is critical to ensure proper drying and thus optimal filament properties during the extrusion process. Is this experimentally confirmed? I think it is redundant.
- How do you mix milled PLA with magnetite NPS? Do you use any protocol to prevent or lower already present agglomeration of MNPs?
- Line 296- The scaffolds were 3D printed using consistent parameters and embedded in an agarose matrix for magnetic hyperthermia measurements, following the experimental protocol previously established for evaluating the heating efficiency of magnetic scaffolds. Each scaffold was subsequently embedded in an agarose matrix for the magnetic hyperthermia measurements, using the same methodology as described previously.
The second sentence is almost the same. I would like you to shortly describe this protocol. Have you determined SAR of commercial magnetite NPS?
- Figure 1- what are percent given below (h,k,l)?
- Line 342- Notably, shifts in the diffraction peaks of the polymer-magnetite composites, relative to those of pure magnetite, suggest modifications in the lattice structure of the magnetite nanoparticles.
It is unclear to me if you have measured XRD for commercial Fe3O4 nps, do we see its XRD pattern on Figure1? Or is it from database? If it is not the same, as in filaments, I think that the stated sentence should not be there. And if yes, please explain more clear what part of process you call the synthesis?? For how long material is at 200C? This part is a bit unclear…with lots of assumptions…
- Line 358- The strong presence of Fe3O4 diffraction peaks highlights the successful incorporation of magnetic nanoparticles into the PLA matrix- I don’t understand this, pls explain.
- Line 354- characteristic peak of PLA in the composite filaments exhibited a significant reduction in 354 intensity or was even absent in some cases, particularly in all PLA types mixed with 20 355 wt% of magnetite nanoparticles. What do you call the characteristic peak of PLA?? Is it amorphous part below 20°?!
- Lines 411-432 should be somewhere in introduction part, not in the results section.
- Line 452. Error in figure number-- Figure 1 a,b,c
- Lines 520-522 This part is repeated. It is already present few lines above
- Line 570- What this means- the nature and velocity of the phenomenon are determined? Please reformulate the sentence
- Magnetic hyperthermia measurements – have you calculated SAR values? Although there are no adiabatic conditions fulfilled in this setup.
- It is not clear why we take special notice to the temperature range 41-45? If I understood well magnetic hyperthermia related to 3d printing scaffolds has role to enhance the bone regenerating process, not to kill cancer cells? Please correct me if I am wrong.
- Best luck!
Author Response
Referee #2:
Comments and Suggestions for Authors
In the manuscript ‘Composite Magnetic Filaments: From Fabrication to Magnetic Hyperthermia Application’, authors employs 3 d printing and application of magnetic nanomaterials to develop effective magnetic filaments. The introduction part is written really well, giving the state of the art in the field, with the most recent references cited. Research tends to focus on understanding of the correlations between the structure and different materials properties in order to predict the thermal performance of the scaffolds and improve its design. Actually, much of the research is devoted to investigation of mechanical properties of modified PLAs.
Study offers an significant amount of experimental data as it covered investigation on three classes of chemically modified biocompatible polymers derived from PLA, test of two structure shapes - a cylindrical shape and a dog-bone shape and also two different concentrations of magnetite nanoparticles: 10 wt% and 20 wt% were tested. They have tried to correlate the filament fabrication process with the mechanical and thermal performance of the final scaffolds.
To my opinion, the work is interesting, could provide data useful for other researchers in the field and I recommend the publication of this manuscript in MDPI Micromachines journal. However, here are a few notes that authors should go through before publication.
Response: We would like to thank the reviewer for their valuable feedback and constructive comments on our manuscript. We appreciate the positive assessment of our work and the recognition of its contribution to the field. Below, we provide detailed responses to each comment and describe the revisions made to enhance the clarity and quality of our manuscript.
- Line 78- Some error appeared
Response: We appreciate your observation. We have reviewed and corrected the cross reference error in line 78 to ensure clarity.
2. Line 158 Regarding pulverization process you stated that the freezing step, followed by repeated pulverization with intermittent re-cooling, ensured consistent particle size distribution in the resulting powder. Is this assumption or is experimentally confirmed?
Response: We appreciate the reviewer’s question regarding the pulverization process. The freezing step, followed by repeated pulverization, was experimentally validated in our previous work [Makridis et al. Composite magnetic 3D-printing filament fabrication protocol opens new perspectives in magnetic hyperthermia. J Phys D Appl Phys, 2023, 56, 285002. https://doi.org/10.1088/1361-6463/accd01.], cited as ref. 15 in the manuscript, where we detailed our protocol for extruding PLA filaments with nanoparticles. Specifically, we reported that maintaining the material below 0°C improves control over the grinding process, leading to a more uniform particle size distribution. Due to space constraints and refinements suggested by other reviewers, the revised manuscript now states: "A) Pulverization: Commercial PLA filaments (EasyFil, Tough, Premium) were manually cut and milled into fine powder, with intermittent freezing below 0°C to enhance particle uniformity." This sentence retains the essential information while streamlining the Methods section. If the reviewer believes additional clarification is needed, we are happy to further elaborate.
3. Line 170 What this sentence means, what is critical regarding lab drying oven at 70C?- Proper temperature control is critical to ensure proper drying and thus optimal filament properties during the extrusion process. Is this experimentally confirmed? I think it is redundant.
Response: We appreciate the reviewer’s feedback. The drying step at 70°C is essential to remove residual moisture before extrusion, as moisture can negatively impact the filament’s mechanical properties and extrusion quality. While this is a common practice in polymer processing, we have revised the sentence for clarity based on suggestions from both reviewers. The updated text now reads: "C) Drying: The composite was dried at 70°C for 1 hour to remove moisture before extrusion." This revised version maintains the key information while improving clarity and conciseness. We hope this addresses the reviewer’s concern.
4. How do you mix milled PLA with magnetite NPS? Do you use any protocol to prevent or lower already present agglomeration of MNPs?
Response: We appreciate the reviewer's question regarding the mixing process and the control of magnetite nanoparticle (MNP) agglomeration. The preparation of composite filaments follows a well-established protocol [Makridis et al. Composite magnetic 3D-printing filament fabrication protocol opens new perspectives in magnetic hyperthermia. J Phys D Appl Phys, 2023, 56, 285002. https://doi.org/10.1088/1361-6463/accd01.], cited as ref. 15 in the manuscript, that ensures homogeneous dispersion of MNPs within the PLA matrix. We ensure homogeneous MNP dispersion by first grinding PLA into granules (~5 mm) and dry-mixing with MNPs. The mixture undergoes controlled extrusion at 10 rpm, where shear forces aid uniform distribution. An automatic filament winder with a laser micrometer ensures consistency. To further reduce agglomeration, the filaments are reground and re-extruded at least three times, following our established protocol. Finally, they are dried at 70°C for 12 hours to prevent moisture-induced clustering, ensuring uniform composite filaments with reproducible properties.
5. Line 296- The scaffolds were 3D printed using consistent parameters and embedded in an agarose matrix for magnetic hyperthermia measurements, following the experimental protocol previously established for evaluating the heating efficiency of magnetic scaffolds. Each scaffold was subsequently embedded in an agarose matrix for the magnetic hyperthermia measurements, using the same methodology as described previously.
The second sentence is almost the same. I would like you to shortly describe this protocol. Have you determined SAR of commercial magnetite NPS?
Response: We thank the reviewer for the comment. In response, we have revised the text to eliminate redundancy. The scaffolds were 3D printed using consistent parameters and embedded in an agarose matrix to simulate a biological environment for magnetic hyperthermia measurements. We followed an established protocol, where an alternating magnetic field (AMF) was applied, and temperature evolution was recorded to evaluate heating efficiency.
Regarding SAR, we have evaluated the SAR of the composite scaffolds and included a discussion on the differences between Specific Loss Power (SLP), which refers to the magnetic nanoparticles, and SAR, which refers to the heating efficiency of the scaffolds in the context of magnetic hyperthermia. This is why we have presented SAR results of the scaffolds rather than SLP values of the magnetic nanoparticles. We also include new Figure 6 in the article that shows the SAR values of the composite scaffolds. We hope this clarifies the methodology and rationale for our focus on SAR.
6. Figure 1- what are percent given below (h,k,l)?
Response: We thank the reviewer for the comment. The percentages listed below (h,k,l) in Figure 1 represent the relative intensity of the diffraction peaks corresponding to the magnetite diffraction PDF card. Each peak in the diffraction pattern has an intensity that differs from others, reflecting the relative strength of the diffraction. By convention, the strongest peak (3 1 1 for magnetite) is assigned an intensity value of 100, and other peaks are scaled relative to this value. We have updated the figure caption to reflect this explanation:
"Figure 1: Comparative X-ray diffraction pattern for all PLA materials (T. PLA, EF. PLA, and Pr. PLA) with and without magnetite MNPs (10 and 20 wt%). The percentages below the h, k, l values correspond to the relative intensity of each diffraction peak, scaled to the strongest peak, which is assigned an intensity of 100."
We hope this clarifies the presentation of the diffraction pattern and the significance of the intensity values in the figure.
7. Line 342- Notably, shifts in the diffraction peaks of the polymer-magnetite composites, relative to those of pure magnetite, suggest modifications in the lattice structure of the magnetite nanoparticles.
It is unclear to me if you have measured XRD for commercial Fe3O4 nps, do we see its XRD pattern on Figure1? Or is it from database? If it is not the same, as in filaments, I think that the stated sentence should not be there. And if yes, please explain more clear what part of process you call the synthesis?? For how long material is at 200C? This part is a bit unclear…with lots of assumptions…
Response: We thank the reviewer for the comment. To clarify, the magnetite nanoparticles (Fe₃O₄) used in this study were commercial, and the X-ray diffraction (XRD) pattern for these commercial Fe₃O₄ nanoparticles was used as a reference. The diffraction pattern in Figure 1 represents both the PLA composites with magnetite and the commercial Fe₃O₄ nanoparticles. The shifts in the diffraction peaks observed for the PLA-magnetite composites relative to the commercial magnetite suggest potential modifications in the lattice structure of the magnetite nanoparticles upon incorporation into the polymer matrix.
Regarding the synthesis process, the commercial magnetite nanoparticles were directly used, and no additional synthesis or modification of the magnetite was conducted during the preparation of the composites. Therefore, the sentence referring to lattice modifications is related to the interaction of the commercial magnetite nanoparticles with the PLA matrix, not a modification during nanoparticle synthesis.
We have updated the manuscript to ensure this distinction is clear and to address any confusion. Thank you for your valuable feedback.
8. Line 358- The strong presence of Fe3O4 diffraction peaks highlights the successful incorporation of magnetic nanoparticles into the PLA matrix- I don’t understand this, pls explain.
Response: We thank the reviewer for the comment. To clarify, the statement refers to the observation that the characteristic diffraction peaks of Fe₃O₄ (magnetite) appear prominently in the X-ray diffraction (XRD) pattern of the PLA-magnetite composites. This strong presence of Fe₃O₄ peaks indicates that the magnetite nanoparticles were successfully incorporated into the PLA matrix. The XRD technique is sensitive to the crystalline structure of materials, and the presence of well-defined peaks corresponding to Fe₃O₄ suggests that the magnetite nanoparticles maintained their crystalline structure within the PLA composite.
Thus, the sentence emphasizes that the magnetite nanoparticles did not undergo significant degradation or transformation during the preparation process and were effectively integrated into the polymer matrix.
We have updated the manuscript to better explain this point. We thank again the referee for the feedback.
9. Line 354- characteristic peak of PLA in the composite filaments exhibited a significant reduction in 354 intensity or was even absent in some cases, particularly in all PLA types mixed with 20 355 wt% of magnetite nanoparticles. What do you call the characteristic peak of PLA?? Is it amorphous part below 20°?!
Response: We thank the reviewer for the comment and for bringing attention to the characterization of PLA in the composite filaments. The characteristic peak of PLA in the XRD pattern refers to the amorphous region, typically observed as a broad halo below 20° 2θ. This region corresponds to the disordered, non-crystalline phase of PLA, which does not exhibit well-defined diffraction peaks due to its lack of long-range order.
In our study, we observed that as the magnetite nanoparticles were incorporated into the PLA matrix, particularly in the composite filaments with 20 wt% magnetite, the intensity of the amorphous peak corresponding to PLA decreased significantly, or in some cases, it disappeared. This reduction in intensity suggests that the presence of the magnetite nanoparticles likely affected the molecular arrangement of the PLA, potentially disrupting its amorphous structure or influencing the crystallization behavior of the polymer.
The impact of magnetite on the PLA matrix could be explained by several factors, including the potential interactions between the nanoparticles and the polymer chains, which may cause the PLA to adopt a different structural arrangement. Alternatively, the magnetite nanoparticles may act as nucleating agents, promoting the formation of crystalline structures in PLA. However, this observation needs further investigation to clarify the exact mechanism.
We appreciate the reviewer's insight into this matter, and the revised explanation should provide a clearer understanding of the observed changes in the PLA XRD patterns due to the incorporation of magnetite nanoparticles.
10. Lines 411-432 should be somewhere in introduction part, not in the results section.
Response: We thank the reviewer for the comment. We understand the suggestion regarding the placement of the discussion in lines 411-432. However, we believe that the content fits appropriately in the results section as it directly supports the findings presented in the manuscript. These lines discuss a comparison with previous studies and provide context for the observed results, reinforcing the relevance of our findings within the framework of existing research. Therefore, we would prefer to retain this section in the results part of the manuscript.
We hope this explanation clarifies our reasoning, and we are open to further suggestions.
11. Line 452. Error in figure number-- Figure 1 a,b,c
Response: We thank the reviewer for pointing this out. We apologize for the error in the figure number. The correct reference should be to Figure S1a, b, c. We will update the manuscript accordingly to reflect this correction.
12. Lines 520-522 This part is repeated. It is already present few lines above
Response: We thank the reviewer for your observation. We have removed the repeated paragraph as suggested, and the text has been updated accordingly.
13. Line 570- What this means- the nature and velocity of the phenomenon are determined? Please reformulate the sentence
Response: We thank the reviewer for the comment. The sentence "With these analyses, the nature and velocity of the phenomenon are determined, and any differences in each case are interpreted separately" has been removed as suggested.
14. Magnetic hyperthermia measurements – have you calculated SAR values? Although there are no adiabatic conditions fulfilled in this setup.
Response: We thank the reviewer for the comment. As mentioned previously, the SAR values have been calculated and included in the article. A new figure (Figure 6) with SAR bar diagrams has been added to provide a clear representation of the results.
15. It is not clear why we take special notice to the temperature range 41-45? If I understood well magnetic hyperthermia related to 3d printing scaffolds has role to enhance the bone regenerating process, not to kill cancer cells? Please correct me if I am wrong.
Thank you for your comment and for the opportunity to clarify. You are correct in noting that the primary role of the magnetic hyperthermia scaffolds in our study is to enhance the bone regeneration process. However, the scaffolds will also play a crucial role in treating bone cancer, not only by promoting regeneration but also by facilitating the targeted destruction of remaining cancer cells. The temperature range of 41-45°C is specifically chosen because it is within the range where magnetic hyperthermia can effectively induce apoptosis in cancer cells, while avoiding thermal damage to healthy tissues. This temperature range allows the composite scaffolds to serve a dual purpose: as implants for bone regeneration and as a therapeutic tool to treat cancer via hyperthermia. The application of alternating current (AC) magnetic fields will generate localized heating within the scaffolds, effectively driving cancer cells in the affected area toward apoptosis, thus enabling a multitherapeutic approach. We have updated the manuscript to reflect this multifunctional role of the scaffolds more clearly. Thank you again for your insight.
16. Best luck!

Reviewer 3 Report
Comments and Suggestions for Authors
The authors presented the development of composite magnetic filaments for biomedical applications, focusing on their synthesis, mechanical behavior, and thermal performance in magnetic hyperthermia. It highlights the use of additive manufacturing techniques to create these filaments, particularly for bone tissue engineering scaffolds. The study evaluates three types of biocompatible polymers derived from polylactic acid (PLA), incorporating magnetite nanoparticles at varying concentrations (10% and 20% by weight). These combinations aim to optimize structure-property relationships, balancing porosity and mechanical strength to replicate the characteristics of natural bone tissue. Experimental results reveal how the polymer type and nanoparticle content significantly influence the performance metrics of the materials. Specifically, the findings indicate that increasing the magnetite concentration enhances the heating efficiency during magnetic hyperthermia treatments. The paper also emphasizes the importance of systematic comparisons between magnetic and non-magnetic counterparts to assess functional properties. The proposed methodologies facilitate the tunability of both mechanical and thermal behaviors of the scaffolds. Ultimately, this research provides critical insights into the design of magnetic scaffolds for targeted tissue regeneration and therapies. The implications extend not only to improved medical applications but also pave the way for advancements in 3D printing and nanocomposite materials.
Several points need to be clarified:
The article does not provide data on the parameters of the magnetic nanoparticles used.
The article would benefit if the authors added drawings, models and photographs of the objects under study
In the section on hyperthermia, it is not clear why in samples with a concentration of 10, 20: the temperature first increases and then decreases.
While the article provides a comprehensive optimization of synthesis parameters for β-FeOOH
After these minor changes, the article can be published.
Author Response
Reviewer 3
Comments and Suggestions for Authors
The authors presented the development of composite magnetic filaments for biomedical applications, focusing on their synthesis, mechanical behavior, and thermal performance in magnetic hyperthermia. It highlights the use of additive manufacturing techniques to create these filaments, particularly for bone tissue engineering scaffolds. The study evaluates three types of biocompatible polymers derived from polylactic acid (PLA), incorporating magnetite nanoparticles at varying concentrations (10% and 20% by weight). These combinations aim to optimize structure-property relationships, balancing porosity and mechanical strength to replicate the characteristics of natural bone tissue. Experimental results reveal how the polymer type and nanoparticle content significantly influence the performance metrics of the materials. Specifically, the findings indicate that increasing the magnetite concentration enhances the heating efficiency during magnetic hyperthermia treatments. The paper also emphasizes the importance of systematic comparisons between magnetic and non-magnetic counterparts to assess functional properties. The proposed methodologies facilitate the tunability of both mechanical and thermal behaviors of the scaffolds. Ultimately, this research provides critical insights into the design of magnetic scaffolds for targeted tissue regeneration and therapies. The implications extend not only to improved medical applications but also pave the way for advancements in 3D printing and nanocomposite materials.
Response: We would like to thank the reviewer for their thoughtful and positive feedback on our manuscript. We greatly appreciate the recognition of the significance of our work in developing composite magnetic filaments for biomedical applications, particularly in the context of bone tissue engineering.
We are glad that the reviewer acknowledges our focus on the synthesis, mechanical behavior, and thermal performance of these filaments in magnetic hyperthermia treatments. The incorporation of magnetite nanoparticles into various biocompatible PLA-based polymers at varying concentrations (10% and 20% by weight) indeed provides a novel approach to optimize the structure-property relationships of the filaments, and we appreciate the reviewer's attention to this aspect.
We are particularly grateful that the reviewer highlighted the importance of our study in demonstrating how the polymer type and nanoparticle content influence the heating efficiency during magnetic hyperthermia treatments. We believe this is one of the key contributions of our work. Additionally, we agree with the reviewer that the systematic comparisons between magnetic and non-magnetic counterparts are essential for understanding the functional properties of the materials, and we have made sure these comparisons are thoroughly discussed in the manuscript.
Finally, we appreciate the reviewer’s acknowledgment of the potential broader impact of our research, not only for targeted tissue regeneration but also for advancements in 3D printing and nanocomposite materials. We believe our findings provide a solid foundation for future work in these areas and can contribute to the development of multifunctional scaffolds for medical applications.
We are grateful for the reviewer's positive and constructive comments, and we will continue refining the manuscript based on this feedback to ensure clarity and accuracy.
Several points need to be clarified:
The article does not provide data on the parameters of the magnetic nanoparticles used.
Response::
We would like to thank the reviewer for pointing out the lack of explicit data on the magnetic nanoparticles used in our study. As mentioned in the Materials and Methods section, the magnetite nanoparticles used in our work are commercial products sourced from Alfa Aesar. We have provided the key specifications of these nanoparticles in the Materials and Methods section, which includes important parameters such as size, shape, and other relevant characteristics. We hope this addresses the concern, and we are happy to provide further clarification or details if needed.
The article would benefit if the authors added drawings, models and photographs of the objects under study
Response:
We appreciate the reviewer’s suggestion to include additional drawings, models, and photographs of the objects under study. We have already included photographs of the scaffolds in Figure 5, which showcase their appearance. Additionally, we have provided photographs of the 3D-printed dog bones and magnetic scaffolds in the Supporting Information (Figure S2). Figure S2 illustrates (a) the 3D-printed magnetic dog bone specimens and (b) the magnetic scaffolds. For each filament used, three dog bones and three magnetic scaffolds were printed to ensure reproducibility and standard deviation in our final results. We believe these additions effectively illustrate the objects under study, addressing the reviewer’s request. If further visual aids or clarifications are needed, we are happy to provide them.
In the section on hyperthermia, it is not clear why in samples with a concentration of 10, 20: the temperature first increases and then decreases.
Response:
We appreciate the reviewer’s comment. To clarify, the observed behavior in the temperature profile for samples with 10% and 20% magnetite concentration, where the temperature first increases and then decreases, is due to the alternating application of the AC magnetic field. In the updated version of Figure 5, we have added dotted lines to indicate the periods when the magnetic field is turned off.
While the article provides a comprehensive optimization of synthesis parameters for β-FeOOH
Response:
Thank you for your comment. While our article focuses primarily on the development of composite magnetic filaments using magnetite nanoparticles, we acknowledge that β-FeOOH is a precursor often involved in the synthesis of magnetite (Fe₃Oâ‚„). However, as the primary material used in our experiments is commercial magnetite, the optimization of synthesis parameters for β-FeOOH was not the focus of our study. If needed, we can clarify the role of β-FeOOH in the materials discussed, specifically in relation to the magnetite nanoparticles used.
After these minor changes, the article can be published.

Round 2
Reviewer 1 Report
Comments and Suggestions for Authors
The authors have improved the manuscript according to the reviewers comments.